# MANI-WM: AN INTERACTIVE WORLD MODEL FOR REAL-ROBOT MANIPULATION

## ABSTRACT

Scalable robot learning in the real world is limited by the cost and safety issues of real robots. In addition, rolling out robot trajectories in the real world can be time-consuming and labor-intensive. In this paper, we propose to learn an interactive world model for robot manipulation as an alternative. We present a novel method, Mani-WM, which leverages the power of generative models to generate realistic videos of a robot arm executing a given action trajectory, starting from an initial given frame. Mani-WM employs a novel frame-level conditioning technique to ensure precise alignment between actions and video frames and leverages a diffusion transformer for high-quality video generation. To validate the effectiveness of Mani-WM, we perform extensive experiments on four challenging real-robot datasets. Results show that Mani-WM outperforms all the comparing baseline methods and is more preferable in human evaluations. We further showcase the flexible action controllability of Mani-WM by controlling the virtual robots in datasets with trajectories 1) predicted by an autonomous policy and 2) collected by a keyboard or VR controller. Finally, we combine Mani-WM with model-based planning to showcase its usefulness on real-robot manipulation tasks. We hope that Mani-WM can serve as an effective and scalable approach to enhance robot learning in the real world. To promote research on manipulation world models, we open-source the code at `https://anonymous.4open.science/r/Mani-WM`.

## 1 INTRODUCTION

The field of embodied AI has witnessed remarkable progress in recent years. Real robots are now able to complete a wide variety of manipulation tasks (Zitkovich et al., 2023). However, real robots are costly, unsafe, and require regular maintenance which may restrict scalable learning in the real world. And rolling out robot trajectories in the real world can be time-consuming and labor-intensive, although it is necessary for model evaluation. While efforts have been made to create powerful physical simulators (Mittal et al., 2023; Chen et al., 2024), they are still not visually realistic enough. Additionally, they are not scalable because building new environments in simulation requires significant effort. What if we can create an interactive world model that simulates robot trajectories in a way that is accurate and visually indistinguishable from the real world? With such a model, agents can interactively control virtual robots to manipulate diverse objects in various scenes and perform model-based planning by imagining the outcomes of different proposed candidate trajectories.

Recent advances in generative models showcase extraordinary performance in generating realistic texts (Achiam et al., 2023), images (Rombach et al., 2022), and videos (Brooks et al., 2024). Inspired by these successes, we propose to leverage generative models in building an interactive world model for robot manipulation in the real world. To this end, we propose Mani-WM, a novel method that generates high-fidelity videos of a robot executing an action trajectory, starting from a given initial frame (Fig. 1). We refer to this task as the *trajectory-to-video* task. The trajectory-to-video task differs from the general text-to-video task in several ways. While various videos can meet the text condition in the text-to-video task, the predicted video in our trajectory-to-video task must strictly and accurately follow the input trajectory. More importantly, a challenge of this task is that each action in the trajectory provides an exact description of the robot's movement in each frame. This contrasts with the text-to-video task, where textual descriptions offer a general condition without specific frame-by-frame details. Another challenge is that the trajectory-to-video task features rich robot-object interactions, which must adhere to physical laws. For instance, when the robot picks

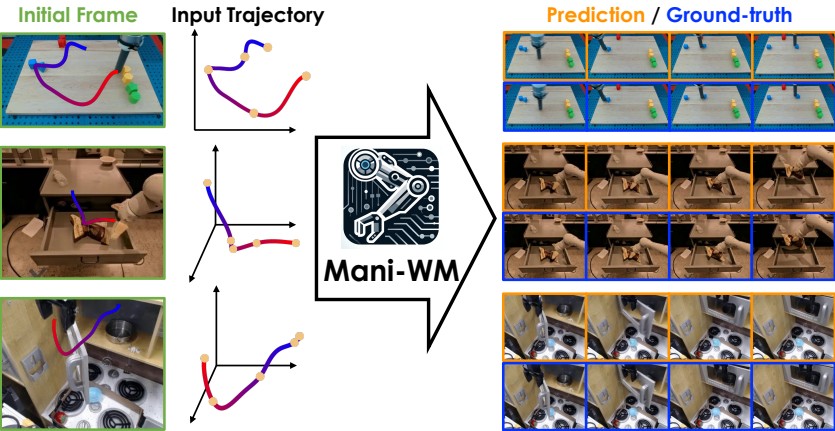

Figure 1: **Overview of Mani-WM.** Mani-WM is an interactive world model for robot manipulation that allows users to input an action trajectory to control the "real robot" in an initial frame.

up a bowl and moves, the bowl should move together with the robot. In terms of data, training a trajectory-to-video model only requires trajectory-video pairs, which is very scalable – even failure trajectories can be used for training.

To tackle the trajectory-to-video task, Mani-WM leverages an innovative frame-level conditioning method to achieve precise frame-by-frame alignment between actions and video frames. We use the powerful Diffusion Transformer (Peebles & Xie, 2023) as the backbone to improve the modeling of robot-object interactions for better compliance with physical laws. To generate long-horizon videos, Mani-WM can be rolled out in an autoregressive manner and maintain consistency between the generated video clips. We validate Mani-WM on four real-robot manipulation datasets: RT-1 (Brohan et al., 2023), Bridge (Walke et al., 2023), Language-Table (Lynch et al., 2023), and RoboNet (Dasari et al., 2020). Results show that Mani-WM can generate high-resolution (up to $288 \times 512$) and long-horizon videos (up to 150+ frames). Compared to baseline methods, Mani-WM achieves superior performance and is more preferable in human evaluations. Moreover, we showcase that Mani-WM is able to generate accurate and realistic videos from trajectories outputted by a policy or collected by humans with a keyboard or VR controller, indicating great flexibility and robustness in real-world application. Finally, we perform model-based plannning experiments on real-robot manipluation tasks with Mani-WM. Results indicate that Mani-WM can accurately imagine the visual outcomes of different proposed candidate trajectories, allowing a model-based policy to select correct trajectories for accomplishing multiple tasks. Please see our project page for videos. To summarize, the contribution of this paper is threefold:

- We propose Mani-WM, a novel method that is capable of generating high-resolution and long-horizon videos for the trajectory-to-video task. It achieves precise alignments between actions and video frames and adheres to physical laws.

- We perform extensive experiments on the trajectory-to-video task with four challenging real-robot datasets. Results show that Mani-WM outperforms all the comparing baseline methods and is more preferable in human evaluations.

- We validate the usefulness of Mani-WM in the real world by conducting real-robot experiments on manipulation tasks. Results show that Mani-WM significantly improves success rates by enabling the policy to foresight the visual outcomes of different candidate trajectories.

## 2 RELATED WORK

**World Models.** Learning a world model (or dynamics model) (LeCun, 2022; Ha & Schmid-huber, 2018), which predicts future observations based on current observations and actions, has recently become increasingly popular (Tian et al., 2023; Hu et al., 2023; Bruce et al., 2024). Prior works (Babaeizadeh et al., 2021; Gupta et al., 2023) train action-conditioned video prediction models for planning on BAIR (Ebert et al., 2017) and RoboNet (Dasari et al., 2020) datasets.

DreamerV3 (Hafner et al., 2023) and DayDreamer (Wu et al., 2023) leverage recurrent state space model (RSSMs) (Hafner et al., 2019) to learn a latent representation of states by modeling a world model for reinforcement learning. iVideoGPT (Wu et al., 2024) trains an autoregressive transformer for action-conditioned video prediction. VLP (Du et al., 2024) exploits text-to-video models as dynamics models to generate video plans for robots. Mani-WM differs from previous works in that it is able to generate high-resolution (up to $288 \times 512$) and long-horizon videos (up to 150+ frames), enabling accurate and flexible world modeling for robot manipulation.

**Video Models.** Video models generate video frames either unconditionally or with conditions including classes, initial frames, texts, strokes, and/or actions (Finn et al., 2016; Ma et al., 2024; Bao et al., 2024; Wang et al., 2024). Recently, diffusion models (Ho et al., 2020) are becoming more and more popular in video generation (Ho et al., 2022; He et al., 2023; Yang et al., 2024; Brooks et al., 2024). A popular choice of architecture is U-Net (Ronneberger et al., 2015) which has also been widely used in image diffusion models (Rombach et al., 2022). Sora (Brooks et al., 2024) showcases extraordinary video generation capability with Diffusion Transformers (Peebles & Xie, 2023). Mani-WM also leverages Diffusion Transformers as the backbone. A relevant line of work is to control video synthesis with motions. These methods use either user-specified strokes (Yin et al., 2023; Chen et al., 2023), bounding boxes (Wang et al., 2024), or human poses (Wang et al., 2023; Xu et al., 2023) as conditions. In contrast, Mani-WM seeks to model complex 3D real-world actions in the video via learning a world model for robot manipulation.

**Scaling Real-World Robot Learning.** Rolling out policies in the real world is essential in scaling up robot learning. Firstly, it is necessary for model evaluation (Zitkovich et al., 2023; Li et al., 2024). Scaling up real-world evaluation would necessitate building and maintaining a large number of robots. To tackle this challenge, recent work (Li et al., 2024) shows a correlation between evaluation in a physical simulator and on real robots. Secondly, as real-robot data are scarce for the reason that data collection often requires costly human demonstrations, an alternative is to roll out a policy to collect data (*e.g.*, dataset augmentation (Ross et al., 2011; Jang et al., 2022; Yu et al., 2023)). Finally, real-robot reinforcement learning requires rolling out robots in the real world to collect trajectories (Levine et al., 2016; Kalashnikov et al., 2018; 2021). However, policy rollout in the real world is time-consuming. And human supervision is often needed to ensure safety which can be labor-intensive. World models are considered a promising solution to these three challenges (Monas & Jang, 2024; Yu et al., 2023; Yang et al., 2024). Our method aims to build a world model for robot manipulation to serve as an efficient and scalable alternative for real-world policy rollout.

# 3 METHODS

## 3.1 PROBLEM STATEMENT

We define the trajectory-to-video generation task as predicting the video of a robot that executes a trajectory given the initial frame $\mathbf{I}^1$ and the action trajectory $\mathbf{a}^{1:N-1}$:

$$\mathbf{I}^{2:N} = f(\mathbf{I}^1, \mathbf{a}^{1:N-1}) \tag{1}$$

where $N$ denotes the number of frames in the video; $\mathbf{a}^i$ denotes the action at the i-th timestep. In this paper, we focus on predicting videos for robot arms. A typical action space for robot arms contains 7 degrees of freedom (DoFs), *i.e.*, 3 DoFs for describing translation in the 3D space, 3 DoFs for 3D rotation, and 1 DoF for the gripper action. The action trajectory $\mathbf{a}^{1:N-1}$ belongs to $\mathbb{R}^{(N-1)\times d}$, where $d$ represents the dimensionality of the action space. Additional details regarding the discussion on the number of context frames and action space are provided in Appendix A.1 & B.

## 3.2 PRELIMINARIES

**Diffusion Models.** Before delving into our method, we briefly review preliminaries of diffusion models (Sohl-Dickstein et al., 2015; Ho et al., 2020). Diffusion models typically consist of a forward process and a reverse process. The forward process gradually adds Gaussian noises to data $\mathbf{x}_0$ over $T$ timesteps. It can be formulated as $q(\mathbf{x}_t|\mathbf{x}_0) = \mathcal{N}(\mathbf{x}_t; \sqrt{\overline{\alpha}_t}\mathbf{x}_0, 1 - \overline{\alpha}_t\mathbf{I})$, where $\mathbf{x}_t$ is the diffused data at the $t$-th diffusion timestep and $\overline{\alpha}_t$ is a constant defined by a variance schedule. The reverse process starts from $\mathbf{x}_T \sim \mathcal{N}(\mathbf{0}, \mathbf{I})$ and gradually remove noises to recover $\mathbf{x}_0$. It can be mathematically

expressed as $p_\theta(\mathbf{x}_{t-1}|\mathbf{x}_t) = \mathcal{N}(\mathbf{x}_{t-1}; \mu_\theta(\mathbf{x}_t, t), \Sigma_\theta(\mathbf{x}_t, t))$, where $\mu_\theta(\cdot)$ and $\Sigma_\theta(\cdot)$ denote the mean and covariance functions, respectively, and can be parameterized via a neural network.

In the training phase, we sample a timestep $t \in [1, T]$ and obtain $\mathbf{x}_t = \sqrt{\overline{\alpha}_t}\mathbf{x}_0 + \sqrt{1 - \overline{\alpha}_t}\epsilon_t$ via the reparameterization trick (Ho et al., 2020) where $\epsilon_t \in \mathcal{N}(\mathbf{0}, \mathbf{I})$. We leverage the simplified training objective to train a noise prediction model $\epsilon_\theta$ as in DDPM (Ho et al., 2020):

$$\mathcal{L}_{\text{simple}}(\theta) = ||\epsilon_\theta(\mathbf{x}_t, t) - \epsilon_t||^2 \tag{2}$$

In the inference phase, we generate $\mathbf{x}_0$ by first sampling $\mathbf{x}_T$ from $\mathcal{N}(\mathbf{0}, \mathbf{I})$ and iteratively compute

$$\mathbf{x}_{t-1} = \frac{\mathbf{x}_t - \sqrt{1 - \alpha_t}\epsilon_\theta(\mathbf{x}_t, t)}{\sqrt{\alpha_t}} \tag{3}$$

until $t = 0$. For conditional diffusion processes, the noise prediction model $\epsilon_\theta$ can be parameterized as $\epsilon_\theta(\mathbf{x}_t, t, \mathbf{c})$ where $\mathbf{c}$ is the condition that controls the generation process. Throughout the paper, we use superscript and subscript to indicate the timestep of a frame in the input video and the diffusion timestep, respectively.

**Latent Diffusion Models.** Directly diffusing the entire video in the pixel space is time-consuming and requires substantial computation to generate long videos with high resolutions (Ho et al., 2022). Inspired by Ma et al. (2024), we perform the diffusion process in a low-dimension latent space $\mathbf{z}$ instead of the pixel space for computation efficiency. Following He et al. (2023), we leverage the pre-trained variational autoencoder (VAE) in SDXL (Podell et al., 2023) to compress each frame $\mathbf{I}^i$ in the video to a latent representation with the VAE encoder $\mathbf{z}^i = \text{Enc}(\mathbf{I}^i)$ where $i \in \{1, 2, ..., N\}$. The latent representation can be decoded back to the pixel space with the VAE decoder $\mathbf{I}^i = \text{Dec}(\mathbf{z}^i)$.

### 3.3 MANI-WM

Mani-WM is a conditional diffusion model operating in the latent space of the VAE introduced in Sec. 3.2. The condition $\mathbf{c}$ consists of the latent representation of the initial frame of a video, $\mathbf{z}^1 = \text{Enc}(\mathbf{I}^1)$, and an action trajectory, $\mathbf{a}^{N-1}$ The diffusion target is the latent representations of the subsequent $N - 1$ frames of the video in which the robot executes the action trajectory, *i.e.* $\mathbf{x} = \mathbf{z}^{2:N}$. Inspired by Sora's remarkable capability of understanding the physical world (Brooks et al., 2024), we similarly adopt Diffusion Transformers (DiT) (Peebles & Xie, 2023) as the backbone of Mani-WM. In the design of Mani-WM, we aim to address three key aspects: 1) consistency with the initial frame 2) adherence to the given action trajectory and 3) computation efficiency. In the following, we describe details of Mani-WM and discuss pivotal design choices to achieve the aforementioned objectives.

**Tokenization.** Each latent representation $\mathbf{z}^i = \text{Enc}(\mathbf{I}^i)$ contains $P$ tokens of $D$ dimensions, where $P$ denotes the number of patches per frame. By sequencing the latent representations of all frames by timestep order, the video is tokenized to $N \times P$ tokens. Spatial and temporal positional embeddings are added to the tokens to allow awareness of patch positions within frames and timesteps in the video, respectively. The VAE is frozen throughout the training process.

**Spatial-Temporal Attention Blocks.** Standard transformer blocks apply Multi-Head Self-Attention (MHA) to all tokens in the input token sequence, resulting in quadratic computation cost. We thus leverage the memory-efficient spatial-temporal attention mechanism (Xu et al., 2020; Bruce et al., 2024; Ma et al., 2024) in the transformer block of Mani-WM to reduce the computation cost (Fig. 2). Specifically, each block consists of a spatial attention block and a temporal attention block. In the spatial attention block, MHA is confined to tokens within a frame to model intra-frame interaction. In the temporal attention block, MHA is confined to tokens at an identical patch position across all the frames to model inter-frame interaction. For a sequence of $N \times P$ tokens, spatial attention operates on the $1 \times P$ tokens within each frame; temporal attention operates on the $N \times 1$ tokens across the $N$ timesteps. Compared to attending over all the $N \times P$ tokens at a time, the spatial-temporal attention greatly decreases the computation cost which makes generating long and high-resolution videos feasible.

**Initial Frame Condition.** The initial frame condition is achieved by treating the initial frame as the ground-truth portion in the input video sequence (Brooks et al., 2024). That is, during training, we

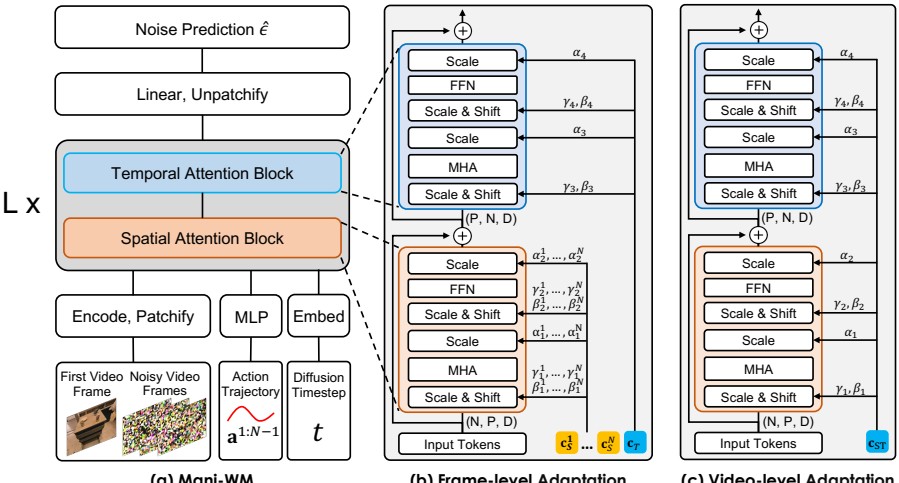

Figure 2: **Network Architecture of Mani-WM**. (a) shows the general diffusion transformer architecture of Mani-WM. The input to Mani-WM includes the initial frame and the given trajectory. (b) Frame-level adaptation (Frame-Ada). (c) Video-level adaptation (Video-Ada).

only add noise to the tokens corresponding to the 2nd to the N-th frames $\mathbf{z}^{2:N}$, while keeping those of the initial frame $\mathbf{z}^1$ intact as it does not need to be predicted (Fig. 2). And the diffusion loss is only computed upon the 2nd to the N-th frames. This condition approach ensures consistency with the initial frame by enabling the predicted frames to interact with it via attention mechanism.

**Trajectory Condition.** A naive approach to impose the trajectory condition is to encode the trajectory as one embedding and append it to the input token sequence as an in-context condition (Peebles & Xie, 2023). However, considering Diffusion Transformers (Peebles & Xie, 2023) demonstrate that adaptive normalization performs better than in-context condition, we adopt this design in Mani-WM to achieve trajectory condition.

- *Video-Level Condition.* Similar to using a text embedding to condition the generation of the entire video in the text-to-video task, we use a linear layer to encode the trajectory into a single embedding for condition. The embedding is then added to the embedding of the diffusion timestep $t$ for generating the scale parameters $\gamma$ and $\alpha$ and the shift parameters $\beta$ for each spatial and temporal attention block. These parameters control the video generation via shifting the distribution of the token embeddings in the transformer block. The overall framework is shown in Figure 2(c). See Appendix C.1 for more details.

- *Frame-Level Condition.* Unlike the text-to-video task where the text describes the entire video, the trajectory in the trajectory-to-video task is a finer description. Each action in the trajectory defines how the robot should move in each frame. And thus, each generated frame must match with its corresponding action in the trajectory. To achieve this precise frame-level alignment, we condition the generation of each frame by its corresponding action. Instead of encoding the action trajectory into a single embedding, we use a linear layer to encode each action into an individual embedding. The diffusion timestep embedding is added to each action embedding to generate the scale and shift parameters for each individual frame in the spatial block. The scale and shift parameters of the temporal block for all frames share the same conditioning embedding which is derived similarly as in video-level condition. See Appendix C.2 for more details.

**Output.** The output layer contains a linear layer which outputs the noise prediction $\hat{\epsilon} = \epsilon_\theta(\mathbf{x}_t, t, \mathbf{c})$. $\hat{\epsilon}$ is used to compute the L2 loss with the ground-truth noise during training (Eq. 2). Note that Mani-WM only predicts the mean of the noise but not the covariance as in Peebles & Xie (2023) – we empirically found that this improves video generation quality. During inference, we sample $\mathbf{x}^T$ from $\mathcal{N}(\mathbf{0}, \mathbf{I})$ and gradually denoise it via Eq. 3 to obtain the predicted latent representation of the 2nd to the N-th frames $\hat{\mathbf{z}}^{2:N} = \mathbf{x}_0$. The predicted video frames can be decoded with the VAE decoder $\hat{\mathbf{I}}^{2:N} = \text{Dec}(\hat{\mathbf{z}}^{2:N})$.

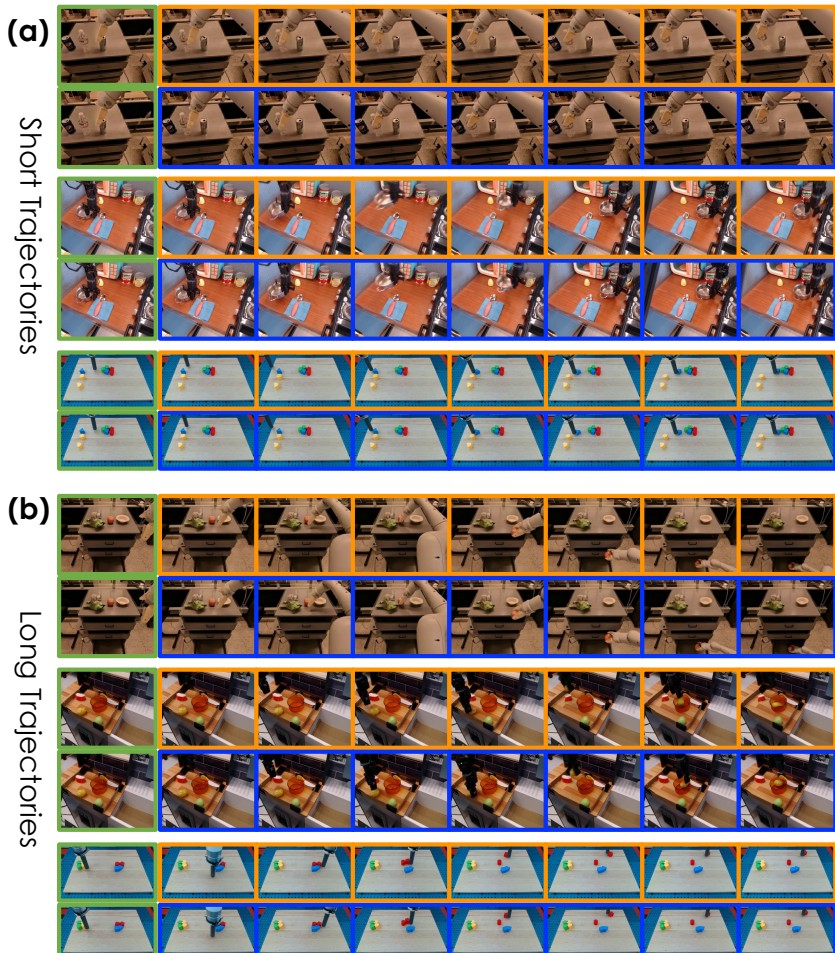

Figure 3: **Qualitative Results.** We show video generation of Mani-WM with (a) short trajectories and (b) long trajectories on the test set of RT-1, Bridge, and Language-Table. Ground-truths are in blue boxes. Predictions are in orange boxes. Initial ground-truth video frames are in green boxes. Please see our project page for videos.

## 4 EXPERIMENTS

In this section, we perform extensive experiments on four challenging real-robot datasets and a real robot. We aim to answer four questions: 1) Is Mani-WM effective on solving the trajectory-to-video task on various datasets with different action spaces? 2) How do different components contribute to the performance of Mani-WM? 3) How is the action controllability of Mani-WM? Can it handle diverse trajectories from humans and policies? 4) Can Mani-WM be used for model-based planning?

### 4.1 EXPERIMENT SETUP

We conduct primary experiments on three high-quality robot manipulation datasets: RT-1 (Brohan et al., 2023), Bridge (Walke et al., 2023), and Language-Table (Lynch et al., 2023). Additionally, we follow iVideoGPT (Wu et al., 2024) to perform experiments on RoboNet (Dasari et al., 2020) to compare with more existing baselines. The action space for RT-1 and Bridge consists of 7 DoF, while the Language-Table features 2 DoF. RobotNet is a mixed dataset with a maximum of 5 DoF. More details about the dataset statistics and action space are shown in the Appendix B. For RT-1, Bridge, and Language-Table during training, we sample video clips containing 16 continuous frames from episodes using a sliding window. For RoboNet, we follow Wu et al. (2024) and use 2 frames as context to predict the next 10 frames. We resize videos, and the resolutions after resizing for RT-1, Bridge, Language-Table and RoboNet are 256×320, 256×320, 288×512 and 256× 256, respectively.

For RT-1, Bridge and Language-Table, we perform experiments on video generation on *short trajectories* and *long trajectories*. Short trajectories, which are segments of complete episodes, consist of 16 frames and 15 actions. The video can be generated in one diffusion generation process. For long trajectories, we utilize complete episodes from the dataset. Long videos can be rolled out in an autoregressive manner. The initial frame of the first diffusion process is the given ground-truth frame, while the initial frame of each subsequent diffusion process is the last output frame from the previous process.

**Mani-WM Variants.** We follow standard transformers which scale the hidden size, number of heads, and number of layers together. In particular, we perform experiments on four configurations: Mani-WM-S, Mani-WM-B, Mani-WM-L, and Mani-WM-XL. Details of these models are shown in Tab. 9 in Appendix E. If not specified otherwise, throughout the paper, we report the results of Mani-WM-XL which contains 679M trainable parameters in total. We denote Mani-WM with frame-level and video-level adaptation as **Mani-WM-Frame-Ada** and **Mani-WM-Video-Ada**, respectively.

**Baselines.** To evaluate the effectiveness of Mani-WM, we first compare it with two state-of-the-art methods, *i.e.*, VDM (Ho et al., 2022) and LVDM (He et al., 2023). Both methods are diffusion models based on a U-Net architecture, in contrast to Mani-WM, which employs a Transformer architecture. LVDM diffuses videos in a latent space, while VDM operates in the pixel space. These methods demonstrate strong capabilities in the text-to-video task. To impose trajectory conditions on video generation, we encode the trajectory into an embedding to condition the diffusion process in both methods. This is similar to the text embedding used for text-to-video generation in the original papers (Ho et al., 2022; He et al., 2023). LVDM is configured such that its number of parameters is similar to Mani-WM. As VDM performs diffusions in the pixel space, it requires more computational resources than LVDM and Mani-WM despite having only 44M parameters. Additionally, we compare Mani-WM with existing non-diffusion methods on the RoboNet dataset, including iVideoGPT (Wu et al., 2024), which autoregressively predicts the next visual token, and MaskViT (Gupta et al., 2023), which generates all visual tokens via iterative refinement. More details can be found in Appendix D.

**Metrics.** Following Xu et al. (2023), we evaluate with two types of metrics: computation-based and model-based. Computation-based metrics includes PSNR (Horé & Ziou, 2010) and SSIM (Wang et al., 2004). Model-based metrics includes Latent L2 loss, FID (Heusel et al., 2017) and FVD (Unterthiner et al., 2019). Unlike the text-to-video task where a variety of videos may meet with a single text condition, the variety is much smaller in the trajectory-to-video task as the robot in the predicted video must strictly follow the input trajectory. Thus, we prioritize the Latent L2 loss and PSNR as primary evaluation metrics and provide other metrics for reference. In Sec. 4.2, we will later show that Latent L2 loss and PSNR match with human preference the most among all the metrics. More details about evaluation can be found in Appendix F.

## 4.2 RESULTS

Table 1: Quantitative Results on Video Generation of Short Trajectories. We prioritize Latent L2 loss and PSNR as primary evaluation metrics.

| Dataset | Method | Computation-based | | Model-based | | |
|---|---|---|---|---|---|---|
| | | **PSNR** ↑ | SSIM ↑ | **Latent L2** ↓ | FID ↓ | FVD ↓ |
| RT-1 | VDM | 13.762 | 0.554 | 0.4983 | 41.23 | 371.13 |
| | LVDM | 25.041 | 0.815 | 0.2244 | **4.26** | 30.72 |
| | Mani-WM-Video-Ada | 25.446 | 0.823 | 0.2191 | 4.34 | 29.27 |
| | **Mani-WM-Frame-Ada** | **26.048** | **0.833** | **0.2099** | 5.60 | 25.58 |
| Bridge | VDM | 18.520 | 0.741 | 0.3709 | 39.82 | 127.25 |
| | LVDM | 23.546 | 0.810 | 0.2155 | 10.59 | 35.06 |
| | Mani-WM-Video-Ada | 24.733 | 0.827 | 0.2021 | **10.30** | 23.03 |
| | **Mani-WM-Frame-Ada** | **25.275** | **0.833** | **0.1947** | 10.51 | **20.91** |
| Language-Table | VDM | 23.067 | 0.857 | 0.3204 | 64.63 | 136.56 |
| | LVDM | 28.254 | **0.889** | 0.1704 | 6.85 | **24.34** |
| | Mani-WM-Video-Ada | 23.893 | 0.859 | 0.2028 | 7.05 | 73.84 |
| | **Mani-WM-Frame-Ada** | **28.818** | 0.888 | **0.1660** | **6.38** | 48.49 |

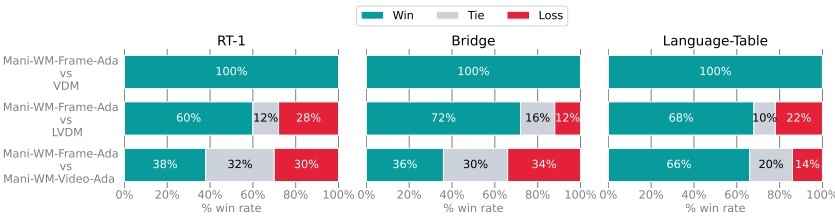

Figure 4: **Human Preference Evaluation.** We perform a user study to evaluate the human preference between Mani-WM-Frame-Ada and other baseline methods.

Table 2: Quantitative Results on Video Generation of Long Trajectories.

| | RT-1 | | Bridge | | Language-Table | |
|---|---|---|---|---|---|---|
| | Latent L2 ↓ | PSNR ↑ | Latent L2 ↓ | PSNR ↑ | Latent L2 ↓ | PSNR ↑ |
| LVDM | 0.2567 | 23.573 | 0.2534 | 21.792 | 0.1776 | 26.215 |
| Mani-WM-Video-Ada | 0.2519 | 23.984 | 0.2385 | 22.868 | 0.2112 | 22.551 |
| **Mani-WM-Frame-Ada** | **0.2408** | **24.615** | **0.2306** | **23.260** | **0.1730** | **26.773** |

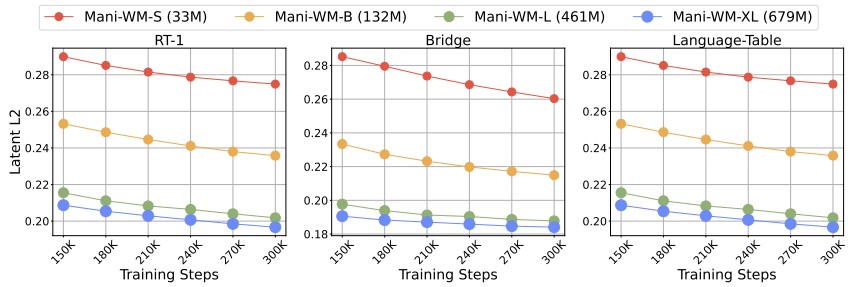

Figure 5: **Scaling.** Mani-WM scales elegantly with the increase of model sizes and training steps.

**Video Generation of Short Trajectories.** Qualitative results are shown in Fig. 3(a), Fig. 8 and Fig. 11. Quantitative results are shown in Tab. 1 and Tab. 3. As shown in Fig. 3(a), Fig. 8 and Fig. 11, Mani-WM-Frame-Ada can generate videos that are almost visually indistinguishable from the ground-truth. As shown in Tab 1, Mani-WM-Frame-Ada performs the best among all the comparing methods in terms of Latent L2 loss and PSNR. It outperforms Mani-WM-Video-Ada in all the computation-based metrics. This indicates that frame-level condition enhances consistency between each frame and its corresponding action in the trajectory, as shown in Fig. 8 in

Table 3: Quantitative Results on RoboNet. * indicates that the result is derived from previous work.

| RoboNet | PSNR ↑ | SSIM ↑ |
|---|---|---|
| MaskViT* | 20.4 | 67.1 |
| iVideoGPT* | 23.8 | 80.8 |
| **Mani-WM** | **24.6** | **81.1** |

the Appendix A.1. Mani-WM-Frame-Ada also surpasses the two baseline methods based on U-Nets on Latent L2 loss. This demonstrates the superiority of transformer-based model, especially in handling complex 3D actions and robot-object interaction. VDM fails to generate realistic videos despite consuming more computation costs during training. This indicates the effectiveness of performing diffusion in latent space. Additionally, as shown in Tab. 3, Mani-WM-Frame-Ada outperforms non-diffusion methods such as iVdeoGPT and MaskViT, demonstrating the superiority of Mani-WM in trajectory-to-video task.

**Human Preference Evaluation.** We also perform a user study to help understand human preferences between Mani-WM-Frame-Ada and other methods. We juxtapose the videos predicted by Mani-WM-Frame-Ada and the comparing method and ask humans which one they prefer. The ground-truth is also provided as a reference. Mani-WM-Frame-Ada beats all the comparing methods in all datasets (Fig. 4). This result aligns with the Latent L2 loss and PSNR which justifies the reason for using them as the primary evaluation metrics. More details can be found in Appendix H.

**Video Generation of Long Trajectories.** Qualitative results are shown in Fig. 3(b) and Fig. 9. Quantitative results are shown in Tab. 2. We compare Mani-WM with the best baseline method LVDM (He et al., 2023). Mani-WM-Frame-Ada consistently outperforms the comparison methods in

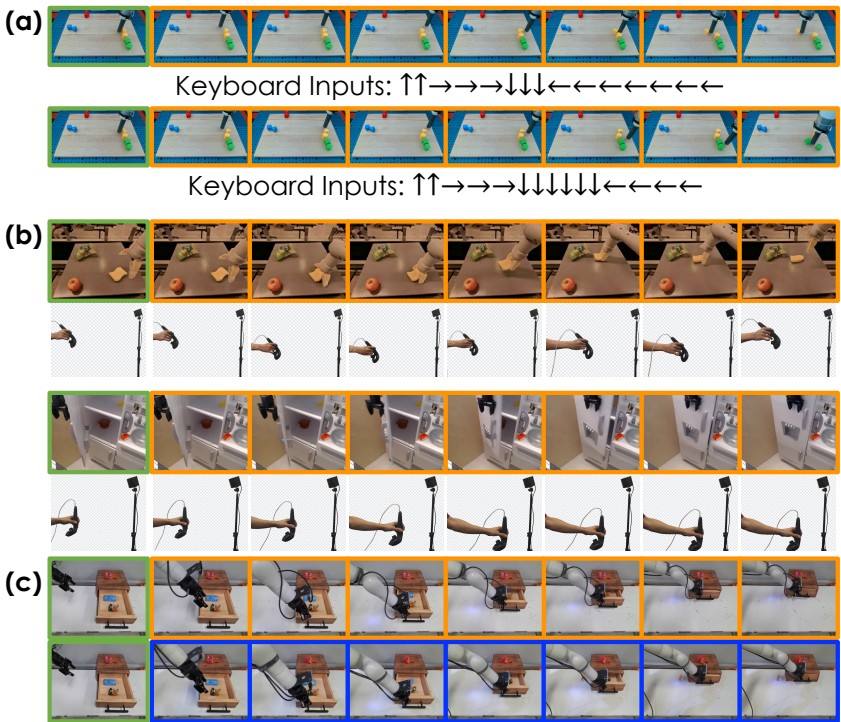

Figure 6: **Flexible Action Controllability.** We showcase controlling (a) the virtual robot in Language-Table with arrow keys on a keyboard, (b) the robots in RT-1 and Bridge with a VR controller, and (c) the robots with a policy. Predictions are in orange boxes. Initial frames are in green boxed. The frames of the real robot execution are in blue boxes.

Table 4: Quantitative results of real-robot model-based planning experiments.

| Method | Close Drawer | Place Mandarin on Green Plate | Place Mandarin on Red Plate | Avg |
|---|---|---|---|---|
| Random | 0.20 | 0.07 | 0.13 | 0.13 |
| Mani-WM (ResNet50) | 0.60 | 0.73 | 0.60 | 0.64 |
| Mani-WM (MSE)) | **0.87** | **0.80** | **0.87** | **0.85** |

all three datasets on Latent L2 loss and PSNR. Fig. 3(b) and Fig. 9 show that it retains the powerful capability of generating visually realistic and accurate videos as in the short trajectory setting.

**Scaling.** We follow Peebles & Xie (2023) and train Mani-WM-Frame-Ada of different model sizes ranging from 33M to 679M. Results are shown in Fig. 5. On all three datasets, Mani-WM scales elegantly with the increase of model sizes and training steps. This indicates strong potential for increasing model sizes and training steps to further improve the performance.

**Flexible Action Controllability.** To showcase the flexible action controllability of Mani-WM, we conduct qualitative experiments in which the virtual robot is guided by trajectories generated from three distinct input sources: a keyboard, a VR controller, and a policy. Importantly, these trajectories exhibit distributions that differ from those in the original dataset. For Language-Table with a 2D translation action space, we use the arrow keys from the keyboard to input action trajectories. For RT-1 and Bridge with a 3D action space, we use a VR controller to collect action trajectories as input. We also train Mani-WM on our own robot dataset and leverage a well-trained policy with action chunk techniques (Chi et al., 2023; Zhao et al., 2023) to predict the trajectories. We compare the video generated by Mani-WM with the corresponding real-robot rollout. Fig. 6 shows that Mani-WM can accurately follow trajectories from different input sources, beyond the training domain. Additionally, Mani-WM is able to robustly handle multimodality in generation. Fig. 6(a) shows videos generated with an identical initial frame but different trajectories. In the Appendix A.4 & A.5, we also demonstrate that Mani-WM can handle noisy and physically implausible trajectories.

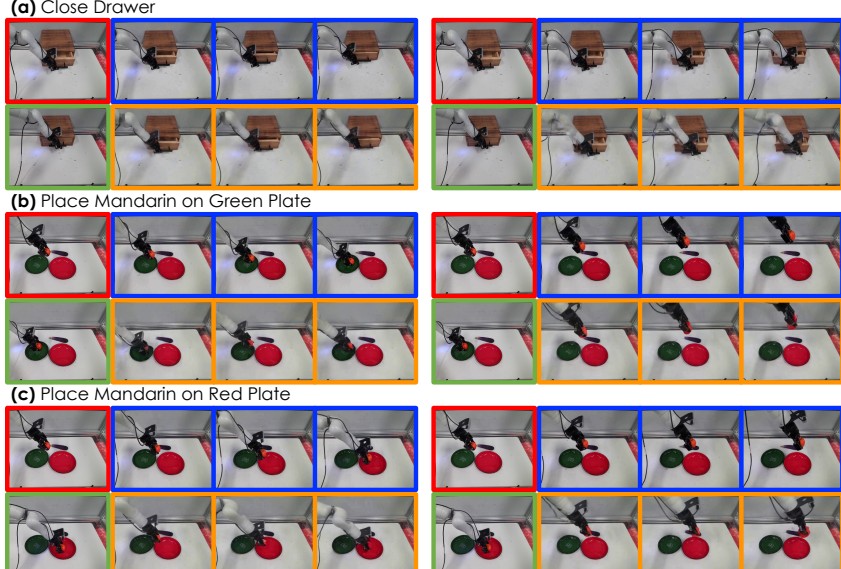

**(a)** Close Drawer

**(b)** Place Mandarin on Green Plate

**(c)** Place Mandarin on Red Plate

Figure 7: Qualitative results of real-robot model-based planning experiments. We conduct experiments across three manipulation tasks and present the rollouts of successful cases (left column) and failed cases (right column). Initial frames are highlighted in red boxes, goal images in green boxes, real-robot rollouts in blue boxes, and predictions made by Mani-WM are displayed in orange boxes.

**Model-based Planning.** We conduct a real-robot model-based planning experiment to show the usefulness of Mani-WM on three manipulation tasks. We leverage a goal-conditioned method which specifies the task with a goal image. In particular, we first sample a set of candidate trajectories. We then use Mani-WM to imagine the visual outcomes of these trajectories and compare them with the goal image via a cost function. The cost function evaluates the similarities between the goal image and a predicted video. The lower the cost, the higher the similarity. The robot rollouts the trajectory with the lowest cost to complete the task. Qualitative results are shown in Fig. 7. Quantitative results are shown in Tab. 4. We experiment with two cost functions for similarity comparison: 1) mean squared error (MSE) and 2) cosine similarity of the feature extracted from ResNet50. We observe that the MSE cost function significantly outperformed the ResNet cost function, and both significantly outperform the policy which randomly selects a trajectory for rollout. These results demonstrate the potential of Mani-WM as a manipulation world model for model-based planning by accurately predicting the visual outcomes of rolling out different trajectories. More details and discussion can be found in the Appendix G.

## 5 CONCLUSION, LIMITATIONS, AND FUTURE WORK

In this paper, we present Mani-WM, a novel method that generates videos of a robot executing an action trajectory given the initial frame. Results show that Mani-WM is able to generate long-horizon and high-resolution videos that are almost visually indistinguishable from ground-truth videos. Additionally, we highlight the flexible action controllability of Mani-WM and its capability for model-based planning.

Similar to other generative models, a limitation of Mani-WM is hallucinations. The hallucinations primarily manifest as violations of physical laws. We believe that an effective way to address this issue is to increase the model size, data volume, and the number of training steps. Additionally, although Mani-WM achieves high throughput with only 8 GB of memory during inference, its inference speed is not real-time. Finally, Mani-WM currently does not support flexible input resolutions, limiting its ability to fully utilize robot data of different resolutions.

In future work, we will investigate accelerating the inference speed using methods such as diffusion distillation (Meng et al., 2023; Ren et al., 2024). Additionally, we plan to explore leveraging Mani-WM as a robot manipulation world model for: 1) policy evaluation (Monas & Jang, 2024); 2) Improving policies via methods such as DAgger (Ross et al., 2011) and RL (Yang et al., 2024).

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

# A    ADDITIONAL QUALITATIVE RESULTS

In this section, we present additional qualitative video results on the following: 1) Short Trajectories: We compare Mani-WM with baseline methods using short trajectories from RT-1, Bridge, and Language-Table. We also provide additional qualitative results of Mani-WM on RoboNet; 2) Long Trajectories: We compare Mani-WM with baseline methods in the long trajectories setting; 3) Scaling: We compare different sizes of Mani-WM; 4) Robustness to Noisy Trajectories: We demonstrate the robustness of Mani-WM when handling noisy trajectories; 5) Robustness to Physically Implausible Trajectories: We show that Mani-WM can handle physically implausible trajectories.

## A.1    VIDEO GENERATION OF SHORT TRAJECTORIES

Qualitative results are illustrated in Fig. 8 and Fig. 11. Fig. 8 demonstrate that Mani-WM-Frame-Ada surpasses other methods in aligning frames with actions and modeling the interaction between robots and objects. For RoboNet dataset, we follow Wu et al. (2024) and use two frames as context for prediction. Fig. 11 illustrates that Mani-WM is capable of simulating the manipulation of flexible objects, such as dragging clothes.

In terms of the number of context frames, we conduct an additional experiment on Bridge dataset and used 2 frames as context. The performance change is minor: the PNSR of using 1 context frame and 2 context frames are both 25. We hypothesize that the input trajectory itself contains sufficient information about velocity. Thus, including more context frames does not bring about significant improvement.

## A.2    VIDEO GENERATION OF LONG TRAJECTORIES

Qualitative results are illustrated in Fig. 9. Mani-WM-Frame-Ada generates consistent and long-horizon videos, accurately simulating the entire trajectory. Additionally, Mani-WM-Frame-Ada maintains its superior performance in frame-action alignment and robot-object interaction as observed in the short trajectory setting.

## A.3    SCALING

Qualitative results are shown in Fig. 10. Mani-WM-Frame-Ada consistently improves the quality of the generated video in terms of reality and accuracy with the increase of model size.

## A.4    ROBUSTNESS TO NOISY TRAJECTORIES

We conduct real-robot experiments to demonstrate Mani-WM's robustness against trajectories with noise. For a trajectory predicted by the policy, we add 5% and 10% Gaussian noise, and we find that Mani-WM is able to handle noisy trajectories robustly, as shown in Fig. 12.

## A.5    ROBUSTNESS TO PHYSICALLY IMPLAUSIBLE TRAJECTORIES

We perform experiments on rolling out a physically implausible trajectory. In particular, we input a trajectory that commands the robot to move downward even after it touches the table. Physically, the robot cannot penetrate the table and thus will remain on the table even if the input control commands it to move down. We input this trajectory to Mani-WM to evaluate its performance in handling physically implausible trajectories. As shown in Fig. 13, Mani-WM can generate physically accurate videos where the robot stays on the table.

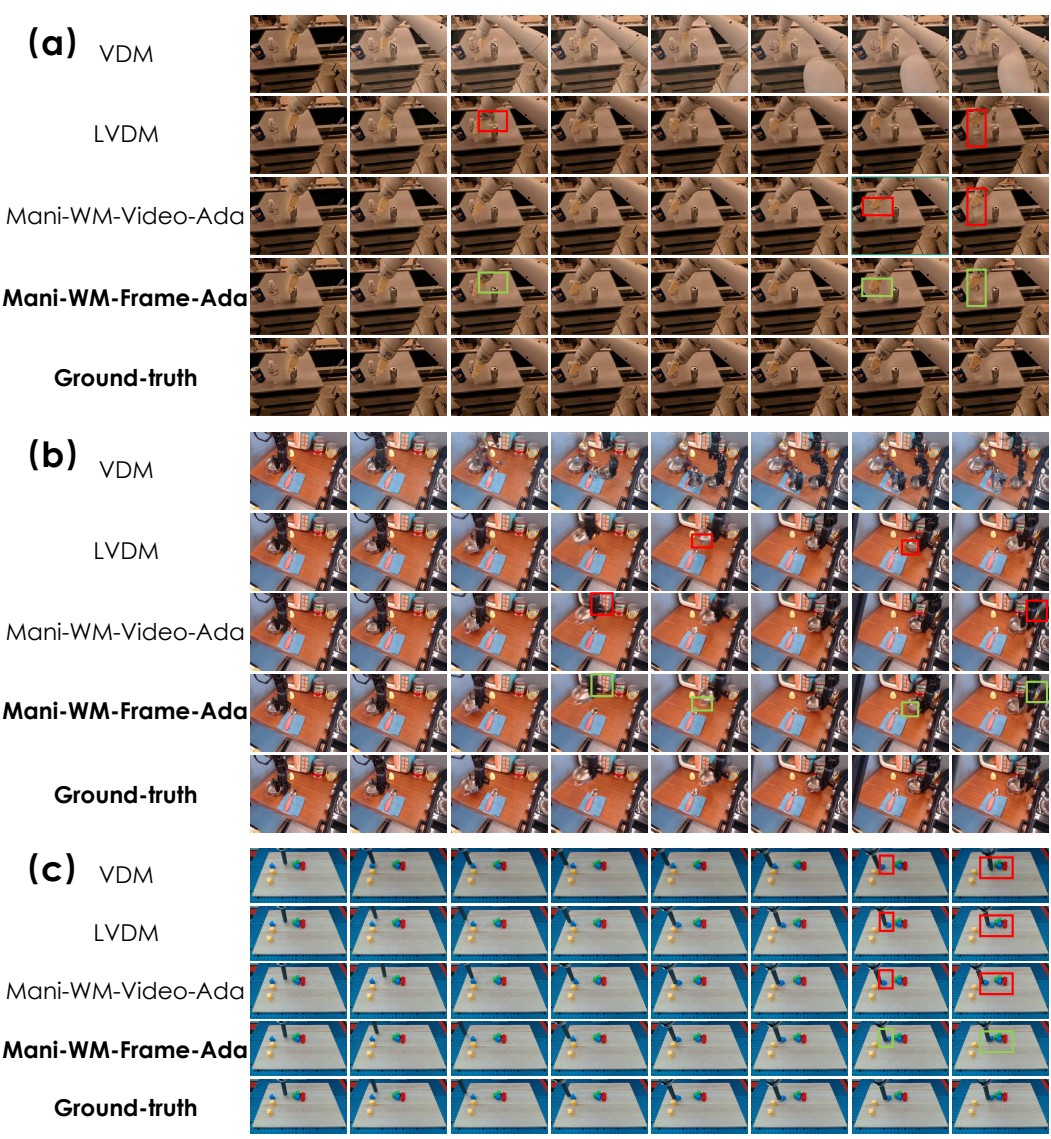

Figure 8: **Additional Qualitative Results on Video Generation of Short Trajectories.** We compare the results of different methods on (a) RT-1, (b) Bridge, and (c) Language-Table. Differences between Mani-WM-Frame-Ada and other methods are highlighted in green and red boxes.

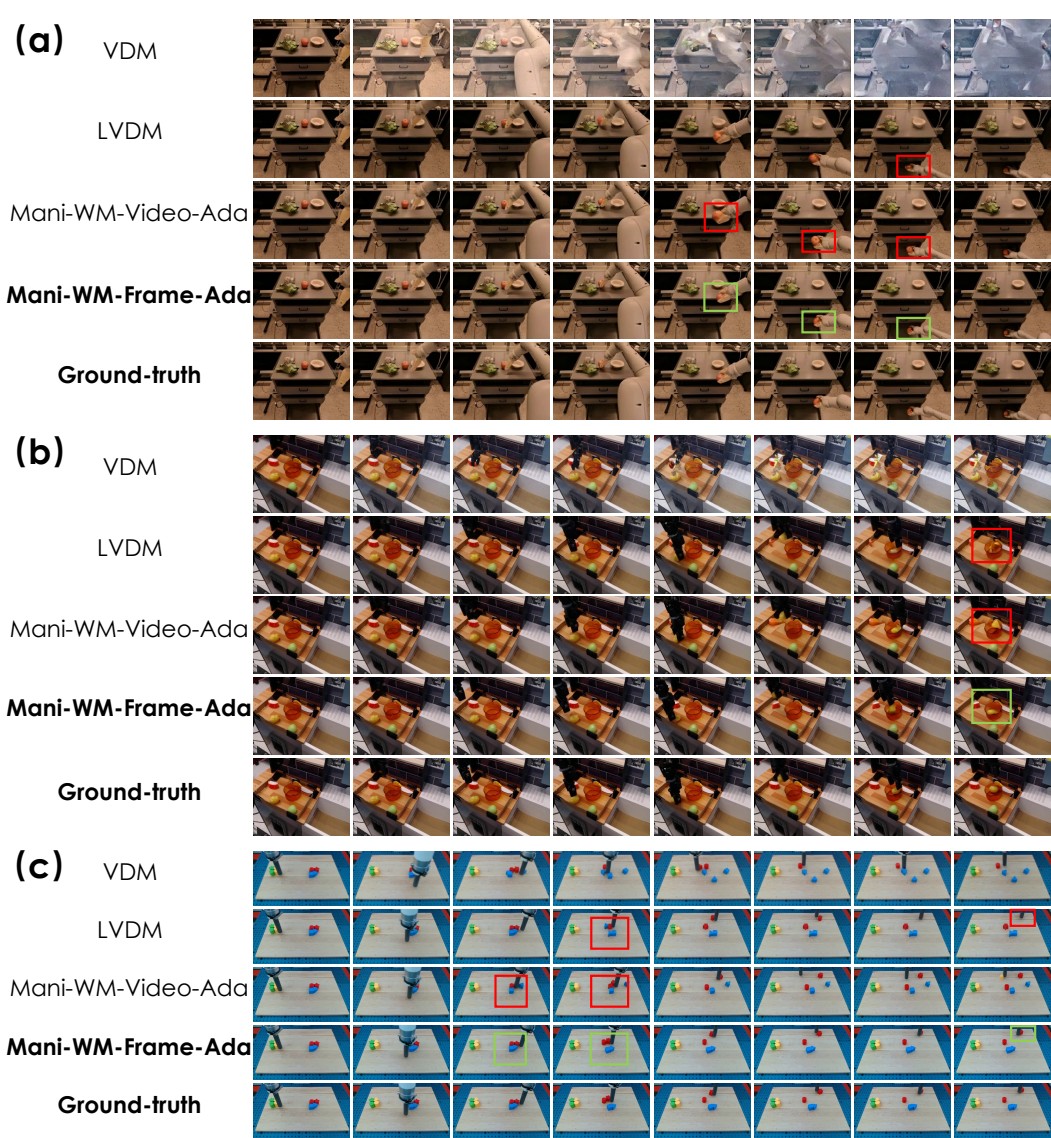

Figure 9: **Additional Qualitative Results on Video Generation of Long Trajectories.** We compare the results of different methods on (a) RT-1, (b) Bridge, and (c) Language-Table. Differences between Mani-WM-Frame-Ada and other methods are highlighted in green and red boxes. Note that the input trajectory is the entire trajectory of an episode.

**(a)** Mani-WM-S

Mani-WM-B

Mani-WM-L

**Mani-WM-XL**

**Ground-truth**

**(b)** Mani-WM-S

Mani-WM-B

Mani-WM-L

**Mani-WM-XL**

**Ground-truth**

**(c)** Mani-WM-S

Mani-WM-B

Mani-WM-L

**Mani-WM-XL**

**Ground-truth**

Figure 10: **Additional Qualitative Results on Scaling.** We compare the results of Mani-WM-Frame-Ada with different model sizes on (a) RT-1, (b) Bridge, and (c) Language-Table.

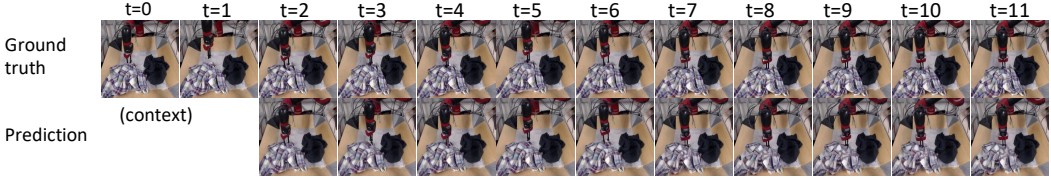

Figure 11: Quantitative results of Mani-WM-Frame-Ada on the RoboNet dataset. The robot is dragging the clothes, indicating that Mani-WM is capable of simulating the deformation of flexible objects.

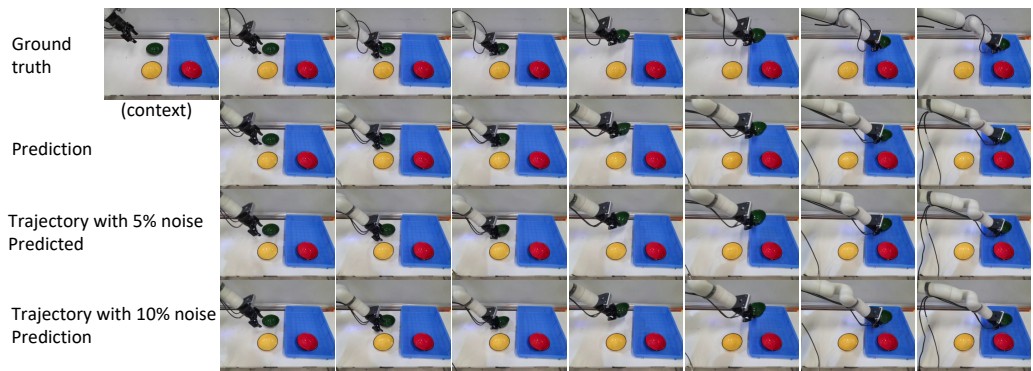

Figure 12: Quantitative results show that Mani-WM is robust to noisy trajectories. The robot is moving a green bowl. We separately present the following in different rows: 1) the trajectory executed by the real robot, 2) the trajectory executed by Mani-WM, 3) the trajectory with 5% noise executed by Mani-WM, and 4) the trajectory with 10% noise executed by Mani-WM. We observe that Mani-WM demonstrates robustness to trajectories with noise.

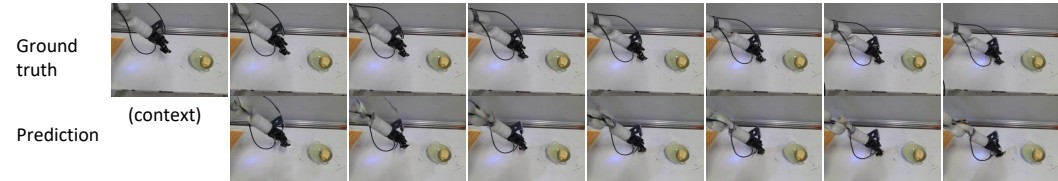

Figure 13: Quantitative results show that Mani-WM is robust to physically implausible trajectories. We control the robot to poke at the table and record the command trajectory, which is very dangerous as it could damage the robot. As a result, the robotic arm is blocked by the table. We find that executing the same trajectory in Mani-WM yields similar results, rather than the robotic arm passing through the table. This indicates that Mani-WM has a certain understanding of the physical laws of the real world.

# B    DATASETS

Table 5: Dataset Statistics. An "episode" is a single trial where the robot completes a task. A "sample" is a clip from an episode. "-" indicates that we follow previous work and do not use a validation set.

| Datasets | RT1 | | Bridge | | Language2Table | | RoboNet | |
|---|---|---|---|---|---|---|---|---|
| Data Split | Episode | Sample | Episode | Sample | Episode | Sample | Episode | Sample |
| Train | 82,069 | 2,314,893 | 25,460 | 482,701 | 170,256 | 1,483,133 | 162,161 | 2,540,500 |
| Validation | 2,167 | 4,810 | 1,737 | 2,905 | 4,446 | 5,119 | - | - |
| Test | 2,167 | 4,799 | 1,738 | 2,946 | 4,562 | 5,243 | 256 | 407 |

**Dataset Statistics.**    We provide details on the four publicly available robot manipulation datasets: RT-1 (Brohan et al., 2023), Bridge (Walke et al., 2023), Language-Table (Lynch et al., 2023) and RoboNet (Dasari et al., 2020). A summary of the dataset statistics is presented in Table 5. For RT-1, Bridge and Language-Table, each training sample consists of a 4-second video clip containing 16 frames, extracted from an episode with a continuous sliding window. For testing and validation, frames are sampled at 16-frame intervals to reduce the number of evaluation videos and, consequently, lower evaluation costs. The original resolution for RT-1 is $256 \times 320$, for Bridge it is $480 \times 640$, and for Language-Table it is $360 \times 640$. To ensure efficient training, we resize the Bridge videos to a resolution of $256 \times 320$ and the Language-Table videos to $288 \times 512$. For RoboNet, we follow Wu et al. (2024) and use 2 frames as context to predict the next 10 frames at a resolution of $256 \times 256$. Note that the mentioned "our own dataset" in Sec. 4.2 is similar in size to RT-1, and the action space is the same.

**Action Space.**    Different datasets have different action spaces. In RT-1 and Bridge, a robot arm with a gripper moves in the 3D space to perform manipulation which interacts with objects in the scene. The action spaces of RT-1 and Bridge consist of 1) 6-DoF arm actions in 3D space, $T \in SE(3)$, and 2) continuous gripper actions, $g \in [0, 1]$. In Language-Table, a robot arm moves in a 2D plane to move blocks with a cylindrical end-effector. The action space of Language-Table is 2-DoF translation in 2D space, $p \in R^2$. We convert the arm action of all datasets to relative delta actions. Specifically, we specify the action of RT-1 and Bridge with a 7-dim vector, i.e., $a = [\Delta x, \Delta y, \Delta z, \Delta\alpha, \Delta\beta, \Delta\gamma, g]$ where $\Delta x$, $\Delta y$, and $\Delta z$ are the delta XYZ position; $\Delta\alpha$, $\Delta\beta$, and $\Delta\gamma$ are the delta Euler angles; $g$ indicates the gripper joint-angle position in the next step. For Language-Table, we specify the action with a 2-dim vector, i.e., $a = [\Delta x, \Delta y]$ which indicates the delta position in the xy-plane. RoboNet is a large-scale robot manipulation dataset featuring 7 robot platforms with varying action spaces (2, 4, or 5 dimensions). Following Dasari et al. (2020), to unify the data, a 5-dimensional vector is used to represent a universal action space, padding zeros for missing dimensions. This vector represents delta XYZ position, delta yaw angle, and gripper joint-angle value: $a = [\Delta x, \Delta y, \Delta z, \Delta\gamma, g]$. For instance, if a robot doesn't control the z-axis, $\Delta z$ is set to 0.

# C    MANI-WM MODEL DETAILS

In this section, we introduce more details about two types of trajectory condition methods in Sec. 3.3: *Video-Level Condition* and *Frame-Level Condition*.

## C.1    VIDEO-LEVEL CONDITIONING

In video-level condition (Fig. 2(c)), we first obtain the conditioning embedding $\mathbf{c}_{ST}$ by adding the diffusion timestep embedding to the trajectory embedding. We then use $\mathbf{c}_{ST}$ to regress the scale parameters $\gamma$ and $\alpha$, as well as the shift parameters $\beta$. Specifically, the computation of the spatial block is as follows:

$$\mathbf{x} = \mathbf{x} + (1 + \alpha_1) \times \text{MHA}(\gamma_1 \times \text{LayerNorm}(\mathbf{x}) + \beta_1) \tag{4}$$

$$\mathbf{x} = \mathbf{x} + (1 + \alpha_2) \times \text{FFN}(\gamma_2 \times \text{LayerNorm}(\mathbf{x}) + \beta_2) \tag{5}$$

where $\mathbf{x}$, with a shape of $(N, P, D)$, denotes the token embeddings. $\mathbf{x}$ is reshaped as $(P, N, D)$ before entering the temporal block. The computation of the temporal block is:

$$\mathbf{x} = \mathbf{x} + (1 + \alpha_3) \times \mathrm{MHA}(\gamma_3 \times \mathrm{LayerNorm}(\mathbf{x}) + \beta_3) \tag{6}$$

$$\mathbf{x} = \mathbf{x} + (1 + \alpha_4) \times \mathrm{FFN}(\gamma_4 \times \mathrm{LayerNorm}(\mathbf{x}) + \beta_4) \tag{7}$$

Note that layer normalization is performed before scaling and shifting.

## C.2 Frame-Level Condition

In frame-level condition (Fig 2(b)), spatial attention blocks and temporal attention blocks are conditioned differently. The derivation of the conditioning embedding for temporal attention blocks $\mathbf{c}_T$ is the same as in video-level condition, where we add the diffusion timestep embedding to the trajectory embedding. Different frames are conditioned differently in spatial attention blocks. We denote the conditioning embedding of spatial attention blocks for the i-th frame as $\mathbf{c}_S^i$. To derive $\mathbf{c}_S^i$, the i-th action in the trajectory is first encoded to an embedding through a linear layer. The diffusion timestep embedding is then added to the encoded embedding to obtain $\mathbf{c}_S^i$. We use $\mathbf{c}_S^1, \ldots, \mathbf{c}_S^N$ and $\mathbf{c}_T$ to regress the corresponding scale parameters $\gamma$ and $\alpha$, as well as the shift parameters $\beta$. While the computation of the temporal blocks is the same as the video-level condition (Eq. 6 and 7), the computation of spatial blocks is different:

$$\mathbf{x}^i = \mathbf{x}^i + (1 + \alpha_1^i) \times \mathrm{MHA}(\gamma_1^i \times \mathrm{LayerNorm}(\mathbf{x}^i + \beta_1^i)), \tag{8}$$

$$\mathbf{x}^i = \mathbf{x}^i + (1 + \alpha_2^i) \times \mathrm{FFN}(\gamma_2^i \times \mathrm{LayerNorm}(\mathbf{x}^i + \beta_2^i)). \tag{9}$$

where $\alpha_1^i, \gamma_1^i, \beta_1^i, \alpha_2^i, \gamma_2^i, \beta_2^i$ denote the scale and shift parameters for the i-th frame. They are regressed from $\mathbf{c}_S^i$.

## D Baselines Details

In this section, we detail the baseline implementation. For VDM (Ho et al., 2022), we leverage the implementation provided in [1], which utilizes a 3D U-Net architecture for controllable video generation. We use only the model component from this code and keep the training setting consistent with Mani-WM. LVDM (He et al., 2023) employs the same model architecture as VDM. It performs diffusion in the latent space while VDM performs diffusion in the pixel space. We use an MLP to encode the trajectory into an embedding. It is then concatenated with the embedding of the diffusion timestep to form the conditioning embedding. This is similar to the original methods in the paper where the text embedding is concatenated with the diffusion timestep embedding to form the conditioning embedding. The initial frame condition method of VDM and LVDM is the same as Mani-WM as described in Sec. 3.3. LVDM and Mani-WM share the same VAE model and training setting. Given that the resolution of Language-Table (Lynch et al., 2023) is up to $288 \times 512$, we resize the video to $144 \times 256$ in the training of VDM to make the computational cost affordable. During evaluation, we resize the generated video back to $288 \times 512$ for comparison with other methods. For RT-1 and Bridge, the training of VDM is performed at a resolution of $256 \times 320$. The training hyperparameters for VDM and LVDM are shown in Tab. 6 and 7. More training hyperparameters that share with Mani-WM can be found in Tab. 8.

We also briefly introduce the baseline details of iVideoGPT (Wu et al., 2024) and MaskViT (Gupta et al., 2023). Both of them use VQGAN (Esser et al., 2021) as the image tokenizer and require additional finetuning it on RoboNet, while Mani-WM employs the VAE encoder from SDXL (Podell et al., 2023) without the need for extra finetuning. Their parameter sizes are 436M and 228M, respectively. Moreover, iVideoGPT undergoes extensive pre-training on OpenX-Embodiment (2023), whereas Mani-WM achieves better video prediction performance with training only on RoboNet.

## E Training Details

For all models, we use AdamW (Kingma & Ba, 2015) for training. We use a constant learning rate of 1e-4 and train for 300k steps with a batch size of 64. The gradient clipping is set to 0.1. We

---

[1] https://github.com/lucidrains/video-diffusion-pytorch

| Table 6: Hyperparameters for VDM. |
| --- |

| Hyperparameter | Value |
| --- | --- |
| Base channels | 64 |
| Channel multipliers | 1,2,4,8 |
| Num attention heads | 8 |
| Attention head dimension | 32 |
| Conditioning embedding dimension | 768 |
| Input channels | 3 |
| Parameters | 40M |

| Table 7: Hyperparameters for LVDM. |
| --- |

| Hyperparameter | Value |
| --- | --- |
| Base channels | 288 |
| Channel multipliers | 1,2,4,8 |
| Num attention heads | 8 |
| Attention head dimension | 32 |
| Conditioning embedding dimension | 768 |
| Input channels | 3 |
| Parameters | 687M |

found the training of Mani-WM very stable – no loss spikes were observed even without gradient clipping. However, loss spikes often occur in LVDM and VDM when gradient clipping is not used. Following Peebles & Xie (2023), we utilize the Exponential Moving Average (EMA) technique with a decay of 0.9999. All other hyperparameters are set the same as Peebles & Xie (2023). Tab. 8 lists further hyperparameters. All models are trained from scratch. We utilize PNDM (Liu et al., 2022) with 50 sampling steps for efficient video generation during evaluation. Mani-WM generates a 16-frame video with a duration of approximately 4 seconds, requiring only 30 seconds on an A100 GPU using 8GB of memory. Although there is still significant room for latency improvement, our method features high throughput and is memory-friendly during inference.

For scaling results in Fig. 5, the configurations of four different sizes of Mani-WM are shown in Tab. 9. We study the scale performance of Mani-WM-Frame-Ada since it performs best.

The information about computing resources for training our Mani-WM is provided in Tab. 10.

## F EVALUATION DETAILS

We introduce the evaluation details in this section.

**Evaluation Metrics.** Latent L2 loss and PSNR measure the L2 distance between the predicted video and the ground-truth video in the latent space and pixel space, respectively. SSIM evaluates the similarity between videos in terms of image brightness, contrast, and structure. FID and FVD assess video quality by analyzing the similarity of video feature distributions.

**Evaluation Setup.** We evaluate the video quality generated by Mani-WM and the baselines under two settings: short trajectories and long trajectories. In the short trajectory setting, the input consists of one initial frame and a short trajectory containing 15 actions, resulting in the generation of 15 subsequent frames. These short trajectories are sampled from episodes using a sliding window with an interval of 16. In the long trajectory setting, the input comprises one initial frame and a complete long trajectory, with the output being the generated subsequent frames. The average lengths of the long trajectories are 42.5, 33.4, and 23.7 frames for RT-1, Bridge, and Language-Table, respectively. These lengths also represent the average number of frames for the generated long videos, which are produced in an autoregressive manner, as detailed in Sec. 4.1. The statistics of the generated short and long videos used for evaluation are presented in Tab. 5.

**Metric Calculation.** In all metric calculations, we ignore the initial frame and only evaluate the quality of the generated frames. For PSNR and SSIM, we refer to skimage [2] for calculation. For FID and FVD, we refer to [3] and [4] for calculation, splitting the generated videos into frames and using their codebases to compute the FID and FVD values. However, we do not calculate FID and FVD metrics for long videos because we find that these metrics do not reflect human preferences well, even in the short trajectory setting. This could be because FID and FVD essentially calculate the similarity

---

[2] https://scikit-image.org/docs/stable/api/skimage.metrics.html

[3] https://github.com/mseitzer/pytorch-fid

[4] https://github.com/universome/stylegan-v

between the distributions of two datasets, whereas the *trajectory-to-video* task is a reconstruction task, making reconstruction loss a more suitable evaluation metric.

Table 8: Hyperparameters for training Mani-WM.

| Hyperparameter | Value |
|---|---|
| Layers | 28 |
| Hidden size | 1152 |
| Num attention heads | 16 |
| Patch size | 2 |
| Input channels | 4 |
| Dropout | 0.1 |
| Optimizer | AdamW($\beta = 0.9, \beta = 0.999$) |
| Learning rate | 0.0001 |
| Batch size | 64 |
| Gradient clip | 0.1 |
| Training steps | 3000000 |
| EMA | 0.9999 |
| Weight decay | 0.0 |
| Prediction target | $\epsilon$ |
| Parameters | 679M |

Table 9: Model Sizes. We use Mani-WM as an abbreviation of Mani-WM-Frame-Ada for brevity.

| Model | Layers | Hidden size | Num attention heads | Parameters |
|---|---|---|---|---|
| Mani-WM-S | 12 | 384 | 6 | 33M |
| Mani-WM-B | 12 | 768 | 12 | 132M |
| Mani-WM-L | 24 | 1024 | 16 | 461M |
| Mani-WM-XL | 28 | 1152 | 16 | 679M |

Table 10: Compution resources for training Mani-WM.

| Dataset | Concurrent GPUs | GPU Hours | GPU type |
|---|---|---|---|
| RT-1 | 32 | 2381 | A800 (40 GB) |
| Bridge | 32 | 2371 | A800 (40 GB) |
| Lanaguge-Table | 32 | 2369 | A100 (80 GB) |

## G  REAL-ROBOT MODEL-BASED PLANNING DETAILS

In this section, we detail the real-robot model-based planning experiment. The experiment demonstrates that Mani-WM can effectively plan trajectories to finish manipulation tasks by generating the outcomes of executing different candidate trajectories.

**Experiment Setup.** We follow Babaeizadeh et al. (2021) to set up this experiment. We implement a model-based policy to show the usefulness of Mani-WM. Our policy consists of a sampling-based planner, a cost function, and Mani-WM as the dynamic function. We first train Mani-WM with our own real robot dataset. The input of our policy includes the initial image, the initial position of the end-effector, and a goal image to indicate the task. The output is a predicted trajectory. We use a simple sampling-based planner to generate candidate trajectories. The planner samples 50 individual points from a circle centered on the initial end-effector position and then generates a trajectory between the initial position and each sampled point, resulting in 50 different candidate trajectories. We input the initial image and each trajectory to Mani-WM to generate the video of executing each trajectory. We use a cost function to calculate the similarity between each predicted video and the goal image. We experiment with 2 cost functions: 1) mean squared error (MSE) and 2)

cosine similarity of the feature extracted from ResNet50. We execute the top 5 trajectories with the lowest cost (i.e., the predicted video most similar to the goal image) in the real world and calculate the average success rate. The experiment is repeated three times for each task.

**Results**  Qualitative results are shown in Fig. 7. Quantitative results are shown in Tab. 4. We compare our method with a baseline that randomly picks a trajectory from the 50 candidates. The results show that using Mani-WM significantly increases the success rate compared to the random baseline.

**Discussion About Cost Function.**  We also explore how different cost functions impact the model's performance. We find that the MSE cost function is generally superior to the ResNet cost function. But the MSE cost function is not always perfect; sometimes it selects incorrect prediction videos, leading to task failure. This suggests that we need to explore better cost functions in future work, considering that the success rate is influenced by both the accuracy of video prediction and the accuracy of the cost function. A suboptimal cost function could affect the evaluation of the video prediction model, as also mentioned by iVideoGPT (Wu et al., 2024) and VLMPC (Zhao et al., 2024).

**Discussion About Sample Policy.**  Although we use a simple sampling-based planner as the sample policy in this experiment, we note that Mani-WM can be combined with any policy that has trajectory sampling capabilities (i.e., action chunk techniques (Chi et al., 2023; Zhao et al., 2023)). The performance and range of tasks that Mani-WM can handle could be further enhanced by adopting a more advanced policy (Chi et al., 2023; Zhao et al., 2023), which is capable of generating more precise and complex trajectories.

## H  HUMAN PREFERENCE EVALUATION

Five participants took part in the human evaluation. For each participant, we randomly sampled 10 ground-truth video clips from the test set for each of the 3 datasets. And for each video clip, we juxtapose the predictions of Mani-WM-Frame-Ada with those of VDM, LVDM, and Mani-WM-Video-Ada (Fig. 14). Thus, a participant evaluated 90 pairs of video clips. Note that the orders of the juxtaposition are random for different clips. See the caption of Fig. 14 for more details. We compare the results of all evaluated video clips and calculate the win, tie, and loss rates. The screenshot of the GUI used in the human evaluation is shown in Fig. 14. The full text of the instruction given to participants is as follows:

---

**Evaluation Instructions**

You are asked to choose the more realistic and accurate video from two generated videos (shown above). The ground-truth video is given as a reference (shown below). Please carefully examine the given videos. If you can find a significant difference between the two generated videos, you may choose which one is better immediately. If not, please replay the videos more times. If you are still not able to find differences, you may choose the "similar" option. Please do not guess. Your decision needs solid evidence.

---

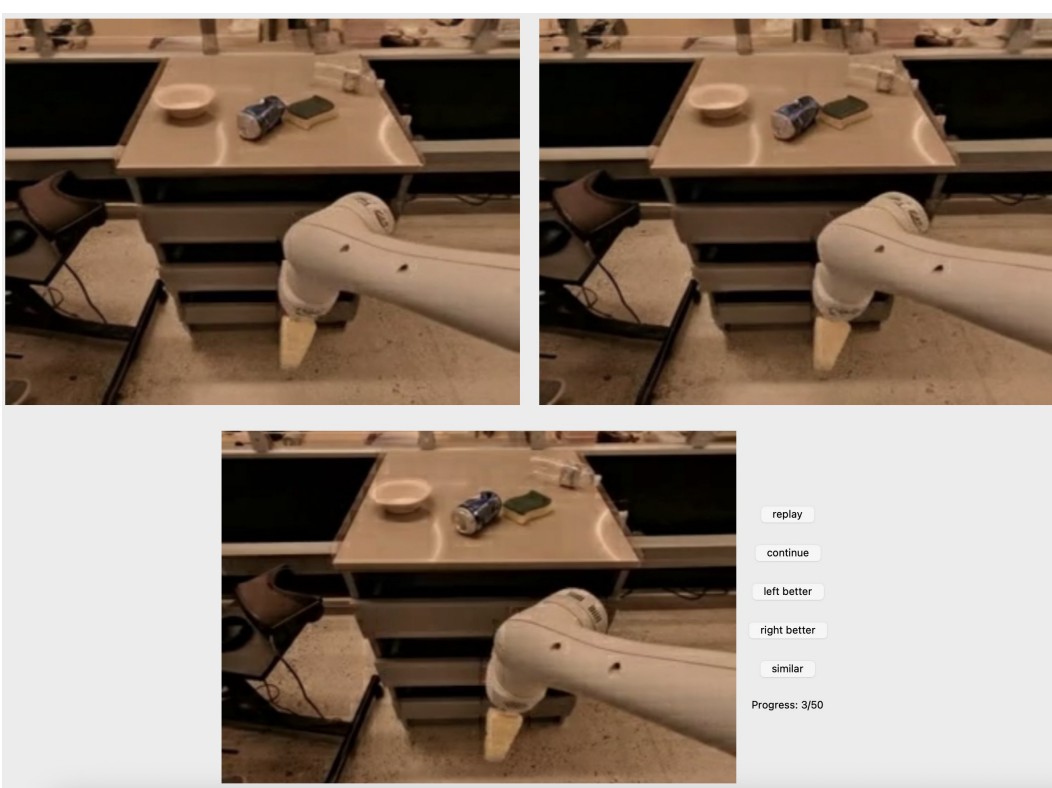

Figure 14: **Screenshot of the GUI in Human Preference Evaluation.** The two videos in the upper row are generated by Mani-WM-Frame-Ada and a comparing method, arranged in a **random** left-right order. The video in the lower row is the ground-truth video.

