# OpenReview forum: "Mani-WM: An Interactive World Model for Real-Robot Manipulation"
_ICLR.cc/2025/Conference — ICLR 2025 Conference Withdrawn Submission_

### Official Review · Reviewer_gtzA · 2024-11-01

**Soundness:** 3
**Presentation:** 3
**Contribution:** 3
**Rating:** 6
**Confidence:** 3

**Summary:**

The paper proposes a new action-conditioned video generation model Mani-WM that is capable of generating a sequence of future images based on the initial image frame and the action trajectory. The proposed model is based on the diffusion transformer and adds a novel frame-level conditioning to incorporate actions. The model is trained on several large datasets separately including RT-1, BRIDGE, Language2Table and RoboNet. It showcased the use of the Mani-WM for predicting the future with policy trajectories and teleoperation commands. They also show how this dynamic model can be combined with model-based planning for real world robotic tasks.

**Strengths:**

1. The paper is well written and explains the framework clearly. It also has very comprehensive appendix to cover the details of dataset, framework and results.
2. Compared to other existing video generation models, the proposed method shows consistent performance gain over different datasets and across different metrics.
3. The paper has done thorough evaluation over different datasets and ablate over the training steps and model sizes.
4. It also showcases the downstream application of policy evaluation, prediction for different controller and model-based sampling.

**Weaknesses:**

1. Since the paper's major contribution is introducing frame-level condition for video generation, it would make the paper stronger to add more discussion about how prior work usually dealt with action and image fusion, even those not for video generation and highlight the benefit of the current conditioning structure.
1. The paper conducts ablation studies on model size and training steps. However, to comprehensively evaluate the scalability of these large models, it would be beneficial to demonstrate how performance scales with dataset size.
2. While the paper presents extensive results on the RT-1, Bridge, and Language-Table datasets, the results for RoboNet are limited. Notably, RoboNet is the only dataset among these that includes multiple embodiments. Therefore, it would strengthen the paper to include more qualitative or quantitative results for RoboNet.
3. For the real robot planning experiments, it is recommended to include other video generation baselines comparisons to better assess performance. Currently, the metrics focus solely on video generation quality when compared to other video generation model. If the ultimate application is planning, it is important to evaluate how improvements in video generation translate to planning performance. For example, planning may prioritize key elements related to task execution rather than every detail in the image. Assessing how failures of capturing dynamics in the baselines affect downstream planning would provide valuable insights for future work. If it is possible, it would be good to show one or two baselines on real robot task in the rebuttal.
4. A minor issue is about the scale of values in some metrics. SSIM is typically reported on a scale from 0 to 1, but in Table 3, both PSNR and SSIM values are scaled by 100. It would improve clarity to explicitly denote this scaling either in the table caption or within the text to ensure consistency and avoid potential confusion for the readers.

**Questions:**

1. For the action-conditioning, the paper mentions it uses delta action as the action space for the trajectory. I am wondering if the proposed framework preprocess the action (e.g., normalization) before feeding into the model.
2. For the section of robustness against the action noise, the paper claims that after adding 5% or 10% gaussian noise to the trajectory, the predicted video is still very similar to the ground truth. However, this seems a little confusing to me. If the learned world model truly captures the dynamics, adding noise to the trajectory should predict a totally different video sequences. I am hoping the authors could add some explanation for this section to clarify potential misunderstanding.
3. For model-based planning, it will help readers understand the results better if the paper can include some details of the customized robot dataset they used for training the world model. Moreover,  it is good to also include whether the tasks and objects used for planning are also presented in the world model training. If they are in the training data for world model, it would be interesting to see how it generalizes to new objects, or new tasks that has similar setup as training data.
4. Regarding using action input from the controller to predict the future images, the paper doesn't mention how they convert those output to the action trajectories. For example, how the up/down arrow key in keyboard is fed into the model? Are they converted to a predefined action step (e.g., 0.01m in y axis)?

---

### Official Review · Reviewer_AVzL · 2024-11-03

**Soundness:** 3
**Presentation:** 3
**Contribution:** 2
**Rating:** 5
**Confidence:** 4

**Summary:**

This paper proposes Mani-WM, a trajectory-conditioned interactive video generation model for robot manipulation tasks. It proposes a novel trajectory-conditioning mechanism that ensures the generated video is aligned with the given trajectory. Experiment results show that Mani-WM outperforms all the comparing baseline methods and is more preferable in human evaluations, and the trained model supports interactive instructions and model-based planning.

**Strengths:**

Strongness:

1.	The video generation results are good, especially on high-resolution videos.

2.	The writing is clear, especially the method section.

3.	The interactive application and the model-based planning demos are good to show the application of this paper.

**Weaknesses:**

Weakness:

1.	To my understanding, the trajectory is to ensure the end-effector in the generated video follows the given trajectory. However, the authors didn’t explicitly evaluate this. Instead, they mainly evaluate the overall generated video quality. This is strange and unfair. So I think: 1) if the authors want to evaluate the quality of the generated videos, they should compare Mani-WM to other video generation models with enhancement techniques, such as video generation with VLM/human feedback, or self-conditioning techniques for diffusion models. Comparing Mani-WM to current baselines is unfair because they are not originally designed for extra trajectory conditions. 2) The author should explicitly evaluate how well the end-effector in the generated videos follow the given trajectory condition.

2.	The authors didn’t explore adding explicit loss to ensure that the generated video meets the input trajectory conditions. For example, the author may consider [1]. Currently, the trajectory conditions are only implicitly injected to the model by training with trajectory conditions. The author may also need to show that videos generated by other generative models satisfy the trajectory condition (even if they do not explicitly use trajectories as input)?

3.	The trajectory condition data acquisition is a limitation of this work. The authors didn’t show how to extract the trajectories of the end-effector from video datasets. This could contain several practical problems: 1) how to deal with the occlusion problem in datasets (the end-effector may be obscured by the objects)? This may cause the trajectory elimination issue, and I think this is very common for long-horizon videos, which is the main type of data that this article mainly promotes; 3) How to deal with stochastic environments dynamics? For example, consider the opening round of billiards. Even if you use the same trajectory every time you tee off, the other balls may not go the same way every time.

4.	Using the end-effector trajectory as the general representation for the condition is not a universal idea for diverse robot video datasets, such as dexterous hand manipulations (where more than one point on the robot are useful) and robot unscrewing the cap (where the end-effector is not moving but rotating). This conflicts with the author's claim that “training a trajectory-to-video model only requires trajectory-video pairs, which is very scalable”.

[1] Geng, Daniel, and Andrew Owens. "Motion guidance: Diffusion-based image editing with differentiable motion estimators." arXiv preprint arXiv:2401.18085 (2024).

**Questions:**

See the weakness.

---

### Official Review · Reviewer_cyei · 2024-11-09

**Soundness:** 2
**Presentation:** 3
**Contribution:** 1
**Rating:** 3
**Confidence:** 4

**Summary:**

This paper proposes Mani-WM, an algorithm for generating high-resolution and long-horizon videos conditioned on robot action trajectories. In terms of video prediction metrics, it is more performance than prior works. It can also be used in the real world for robot manipulation through visual model-based planning.

**Strengths:**

- Paper’s presentation is clear, concise, and easy to follow with nice figures
- The video prediction experiments are extensive
- The results on controlling the robot using various teleoperation methods is interesting, and could potentially open up new pathways for easier data collection. Though I am doubtful about how generalizable this is, see the questions section.

**Weaknesses:**

- I fundamentally do not believe more precise reconstruction or higher resolution leads to better robot learning. Not all pixels have to be correct as long as the video is semantically correct, or as long as the task-relevant object and the robot are correctly predicted. Popular model-based RL methods like DreamerV3 even model the dynamics purely in the latent space and still perform very well and are very data-efficient. I would even argue that, if the downstream use case of a world model is for model-based planning, then generalization to unseen action trajectory is a much more important metric, since most sampled actions will be out of distribution. Since the paper claims that Mani-WM is designed for robot manipulation, I believe it has spent too much experimental efforts in evaluating video prediction metrics and quality, and too little in evaluating actual downstream policy performance. If I have somehow missed the point of the paper, please let me know and I am happy to further discuss this.
- Missing citations/discussions/baselines: UniSim, dreamitate, opensora
- The policy learning section of the experiments is particularly weak and unsubstantial. The authors should include results on using baseline video prediction methods included in previous sections for model based planning too. There is no detail about the data used to train Mani-WM in the real world. To justify using video prediction model for policy learning, the authors should include other policy learning approaches as baseline, such as behavior cloning with diffusion policy, for instance. In addition, to help with the claim of the paper, the authors should perform experiments where Mani-WM is used in other policy learning framework such as model-based RL, as done in iVideoGPT.
- The numerical improvement of Mani-WM over prior works as listed in the quantitative results section seems relatively marginal. I do want to say that I am not familiar with how much improvement is 1 point in PSNR, for instance, so the authors can provide more context
- As a followup to the previous point, in the qualitative visualization section, the authors should include some comparison with baseline methods. This will help provide more context to the Mani-WM’s superior quantitative results and show whether or not it is marginal.

**Questions:**

- What is Mani-WM’s zero-shot generalization capabilities? I believe this is very important for a world model, especially if it is used for model-based planning. How does Mani-WM perform with OOD unseen action trajectories?   The most convincing demonstration of this is to put up an interactive visualization on the website, where the user can specify any desired 7dof, or at least 2d, trajectories, and visualize the generated videos.
- More details on the real robot training dataset, how much data used and are they different from the test time?
- Are the samples used in the qualitative results from the training or validation set?
- Is the model-based planning experiment done open-loop or closed-loop? If it is closed loop, what is the planning frequency?
- The authors claim that Mani-WM is better than previous world model methods because it can predict higher resolution images with longer horizon. But why does higher resolution images matter for robot task performance? And why is longer horizon better for robot task performance when generally closed-loop policies with short horizon and more frequently replanning tend to be more reactive and work better?

---

### Note · Authors · 2024-11-22

**Comment:**

We decide to withdraw the submission and further improve the paper based on comments and suggestions raised by the reviews. We sincerely appreciate all the reviewers for their valuable and constructive feedback and time.

**Withdrawal Confirmation:**

I have read and agree with the venue's withdrawal policy on behalf of myself and my co-authors.